# The Effect of Environmental Factors on the Nutrition of European Beech (*Fagus sylvatica* L.) Varies with Defoliation

**DOI:** 10.3390/plants12010168

**Published:** 2022-12-30

**Authors:** Mladen Ognjenović, Ivan Seletković, Mia Marušić, Mathieu Jonard, Pasi Rautio, Volkmar Timmermann, Melita Perčec Tadić, Miran Lanšćak, Damir Ugarković, Nenad Potočić

**Affiliations:** 1Department of BI, Analytics and Research, Njuškalo Ltd., 10000 Zagreb, Croatia; 2Croatian Forest Research Institute, 10450 Jastrebarsko, Croatia; 3Earth and Life Institute, Université Catholique de Louvain, 1348 Louvain-la-Neuve, Belgium; 4Natural Resources Institute Finland, 00790 Helsinki, Finland; 5Norwegian Institute of Bioeconomy Research, 1433 Ås, Norway; 6Meteorological and Hydrological Service, 10000 Zagreb, Croatia; 7Faculty of Forestry and Wood Technology, University of Zagreb, 10000 Zagreb, Croatia

**Keywords:** tree vitality, foliar composition, stoichiometry, climate change, ICP Forests

## Abstract

Despite being adapted to a wide range of environmental conditions, the vitality of European beech is expected to be significantly affected by the projected effects of climate change, which we attempted to assess with foliar nutrition and crown defoliation, as two different, yet interlinked vitality indicators. Based on 28 beech plots of the ICP Forests Level I network, we set out to investigate the nutritional status of beech in Croatia, the relation of its defoliation and nutrient status, and the effects of environmental factors on this relation. The results indicate a generally satisfactory nutrition of common beech in Croatia. Links between defoliation and nutrition of beech are not very direct or very prominent; differences were observed only in some years and on limited number of plots. However, the applied multinomial logistic regression models show that environmental factors affect the relationship between defoliation and nutrition, as climate and altitude influence the occurrence of differences in foliar nutrition between defoliation categories.

## 1. Introduction

Meteorological and climate conditions affect vegetation directly and indirectly. Direct effects include responses to temperature; indirect effects occur primarily as soil-mediated phenomena, such as the influence of precipitation on soil moisture regimes [1,2,3,4]. Probably the greatest threat to existing forest ecosystems in Europe comes from a changing climate [5]. Extreme climatic events such as droughts are thought to be important in initiating changes in forest ecosystems [6]. According to IPCC [7] predictions, extreme climate events such as heat waves and long-lasting summer droughts will occur more frequently. Such climate changes will negatively impact the stability, structure and biodiversity of forest ecosystems throughout Europe [8]. The region of southeast Europe is one of the most vulnerable regions in Europe when it comes to climate change impacts, primarily through intensified severity and duration of droughts and heat waves. As these impacts should be stronger and faster than on the rest of the continent [9,10] this region is an ideal model for studying the future impact of changing climatic conditions.

A prerequisite for healthy growth and balanced metabolism is the maintenance of adequate concentrations and relatively stable ratios of nutrients in plant tissues [11]. Concentrations of nutrients and their relationships in the leaves of forest trees are important indicators of their functioning. It is crucial to account for nutrient limitation when studying the forest response to climate change [12], as nutritional status has a significant and diverse impact on tree vitality [13]. The concentration of nutrients in leaves depends on many factors: the amount and distribution of precipitation and the length of the growing season, the presence of nutrients in the soil, ion antagonisms, ion mobility and uptake capacity, etc. [14]. It seems that the nutritional status of European beech (*Fagus sylvatica* L.) is deteriorating: Jonard et al. [12] and Talkner et al. [15] both report on the negative trend in beech phosphorus (P) nutrition; furthermore, trends of concentrations of calcium and magnesium in the foliage of beech trees in Europe were also found to be negative [12]. Nutrient contents of tree foliage, which reflect atmospheric and soil-related influences, are an important part of the UNECE ICP Forests monitoring scheme [16]. Beech forests in Croatia have a very wide edaphic amplitude [17], but baseline data on the chemical composition of foliage in beech stands has so far been missing.

Tree vitality is defined as the ability of a tree to assimilate, to survive stress, to react to changing conditions, and to reproduce [18]. As vitality cannot be measured directly, various indicators can be used to describe it [19]. Crown defoliation is a commonly used tree vitality indicator [20,21,22], which can be obtained cost-effectively and relatively quickly in field surveys [23]. Defoliation is defined as the loss of needles or leaves in the assessable part of the crown compared to a reference tree [24]. Trees with defoliation above 25% are regarded as considerably defoliated and are thus defined as damaged [25]. A recent study showed that there are significant morphological and, in this context even more importantly, physiological differences in beech leaves between healthy and considerably defoliated trees [22]. Although European beech had the lowest mean defoliation (20.9%) of all main tree species in the ICP Forests 2021 survey, the 20-year trend in mean plot defoliation shows that defoliation of beech is constantly on the rise (3.4% in 20 years), and that larger deviations from this trend can be linked to drought events [26]. A good example is year 2004, when the annual mean plot defoliation was higher than the trend as a result of the drought in the preceding year which affected large parts of Europe [27,28,29]. In Croatia, the trend in mean defoliation of beech is much steeper with an 8.6 % rise in the period of 1996–2017 [30].

European beech is often mentioned in the context of sensitivity to high temperatures and drought [31,32], as beech grows best in areas marked by moderately warm summers and high quantities of precipitation [33]. Beech forms vast forest communities which provide multiple ecosystem goods and services, and is the dominant tree species in many deciduous forests in Europe under temperate climate conditions [34]. The projected effects of climate change, particularly drought, are expected to significantly affect the vitality of beech, which we attempted to assess with foliar nutrition and crown defoliation, as two different, yet closely interlinked vitality indicators. Although considered a nonspecific indicator of vitality, defoliation appears to depend on the nutritional status of trees. However, research on the relationship between tree defoliation and nutritional status has so far been limited, especially for beech: Jonard et al. [35] found that defoliation was associated with lower concentrations of calcium (Ca) and magnesium (Mg) in beech leaves. A study conducted in Switzerland did not find a link between nitrogen (N) concentrations and defoliation [36], but Ferretti et al. [37] report that the proportion of common beech trees with more than 25% defoliation increases with elevated nitrogen to calcium and potassium (K) ratios in the leaves, indicating that crown status depends on the balance of nutrients in leaves. Toigo et al. [38] found that defoliation increased due to drought effects, while the influence of nutritional status was less consistent. The study by Ognjenović et al. [39] on a limited sample of 26 beech trees showed that defoliation and calcium leaf concentrations show similar susceptibility to drought, but with different responses in time. Obviously, besides the overall tree health condition, the concentration of nutrients in leaves depends on many outside factors. In this context, we set out to better characterize the link between defoliation and foliar nutrition and determine the influence of environmental factors on the foliar composition of beech trees of two distinct defoliation categories, on a large sample of beech-dominated ICP Forests Level I plots in Croatia, with the hypotheses that (I) environmental factors have a significant influence over the defoliation/nutrition relationship and (II) trees with higher defoliation have lower concentrations of nutrients in leaves. In the process, we investigated whether tree nutrition plays a role in the loss of vitality of beech trees (and vice versa); and if the link between tree nutrition and vitality loss varies from year to year due to interannual climate variations.

## 2. Results

### 2.1. Foliar Nutrient Concentrations

Mean N foliar concentrations in the study area level were within normal range [40] in both defoliation categories and across sampling years. Repeated Measures ANOVA indicated that there was no difference in N concentrations between LD and HD trees, while there was a difference between individual years (F(2, 556) = 30.4, *p* < 0.0001). In 2018, significantly lower N concentrations were recorded for both defoliation categories compared to 2019 and 2020 (Table 1).

Although there is no significant main effect of defoliation category on P concentrations, the pairwise comparisons between LD and HD trees indicated significant differences in 2019, when LD trees had a higher concentration than HD trees, but not in 2018 and 2020 (Figure 1). Significant differences in P concentrations (F(2, 556) = 74.9, *p* < 0.001) were also observed between individual sampling years (Table 1) with a decrease during the study period. However, values of P on the plot level were mostly normal throughout the study period except for the year 2020, when P concentrations were slightly below normal range on 53% of the research plots (Appendix A).

No significant differences were established for potassium (K) between defoliation categories and mean concertation values stayed within the normal range in each year (Figure 1).

Repeated Measures ANOVA showed no significant main effect of defoliation category on Mg concentrations. Nevertheless, pairwise comparisons between LD and HD trees revealed significant differences in 2018, with HD trees having higher Mg concentrations (Figure 1). Mean Mg concentrations were within normal range in each year, although they were lower in 2019 and 2020 compared to 2018 for HD trees.

Similar to other elements, there was no significant main effect of defoliation category on calcium (Ca) concentrations (F(2, 556) = 3.4, *p* = 0.06). Pairwise comparisons, however, showed that HD trees in 2018 and 2019 had significantly higher Ca concentrations than LD trees (Figure 1). For these years, values for both groups fall within normal Ca concentration range for beech. In 2020, both categories had significantly higher concentrations than in previous years, bordering on surplus (luxury) values, but no significant difference was found between them (Figure 1).

Although there was no significant main effect of defoliation category on dry leaf mass, the pairwise comparisons between LD and HD trees indicated significant differences in 2018, when LD trees had a higher dry leaf mass than HD trees (Figure 1). However, there was a statistically significant main effect of the sampling year on dry leaf mass (F(2, 556) = 56.3, *p* < 0.001) and pairwise comparisons showed differences between all years for both defoliation categories (Table 1).

On the plot level, mean Ca concentrations were mostly in the normal range during the first two years, while excessive Ca concentrations were observed on all plots located in the alpine region in 2020 (Appendix A), and most of the alpine region plots had insufficient P concentrations (Appendix A). In the same year, excessive K concentrations were found on eleven plots (39%), most of them located in the continental region (Appendix A). N values are rather plot dependent and relatively stable in time (Appendix A), while Mg concentrations show a very broad range of values, independent of the area, defoliation category, or sampling year (Appendix A)

Overall, the share of plots with differences in foliar concentrations between defoliation categories was relatively low for most elements during the entire sampling period (Table 2). The highest share of plots with differences in foliar concentrations between defoliation categories was determined for Ca (21.4%) in 2018, followed by Mg (17.9%) in the same year.

### 2.2. Foliar Nutrient Ratios

N/P ratios were especially high and above the upper limit for both LD and HD trees in 2020 at the study area level, mostly due to very low P values that approached deficiency values (Figure 2). Repeated Measures ANOVA did not show a significant main effect of defoliation category on N/P ratio. However, pairwise comparison between LD and HD trees indicated significant differences in N/P ratios in 2019, with values out of optimal range for HD trees due to low P concentration values (Figure 2). The main effect of the sampling year was significant (F(2556) = 13.4, *p* < 0.001) with an increase in N/P ratio during the monitoring period.

No significant differences were found for N/K ratio between defoliation groups or sampling years and values were within normal range in all years.

Repeated Measures ANOVA identified a significant main effect of defoliation categories on N/Ca ratios (F(2556) = 109.9, *p* < 0.0001). Pairwise comparisons confirmed differences in N/Ca ratios between defoliation groups in 2018 and 2019 (Figure 2). These differences were dictated primarily by Ca concentrations, which showed higher values in HD trees in 2018 and 2019. Sampling year also had a significant effect (F(2556) = 13.4, *p* < 0.001) and pairwise comparison established that values within defoliation categories were lower in 2020 compared to 2018 and 2019 (Table 3).

Due to low Mg concentrations in 2019 and 2020, the N/Mg ratios were mostly in the surplus range in those years (Figure 2). No significant effect of defoliation categories on N/Mg ratios was found. However, pairwise comparisons showed a significant difference between LD and HD trees in 2018, with higher values for LD trees.

Mean N/P, N/K and N/Ca ratios on the plot level were mostly in normal range during the entire study (Appendix A). However, in 2020, most of the plots located in the alpine biogeographic region had excessive N/P and insufficient N/Ca ratios.

The highest share of plots with differences between defoliation categories (5 out of 28, or 17.9%) was determined for N/Ca in 2018 and N/P in 2019. The share of plots in which we found differences hardly passed 10% for other ratios and years (Table 4).

### 2.3. Effects of Environmental Factors on Differences in Foliar Nutrition

While the fitted MLR (multinomial logistic regression) models for N and P did not indicate a significant influence of any of the tested factors, we found that environmental factors influence the occurrence of differences in foliar concentrations of K, Ca, and Mg between defoliation categories.

Altitude (alt) and mean annual temperature (MAT) are the only environmental factors that influence the occurrence of Differences in Foliar Concentrations (DFC) of K. Keeping all other variables constant, the odds for LD trees having higher K concentration is 0.5% higher for every meter increase in altitude. On the other hand, odds are 35% lower for HD trees to have higher K concentrations when the mean temperature increases.

Trees on higher altitudes are more likely to have higher Ca concentrations in HD trees (Table 5). An opposite effect can be seen with mean annual precipitation (MAP) where, keeping all other variables constant, with an increase in precipitation the odds are 90% higher for LD trees and 2% lower for HD trees to have higher Ca concentrations. With increasing maximum temperatures (Tmax) it is 1.3 times likely that HD trees will have significantly higher Ca concentrations.

Beech trees on higher altitudes are also more likely to have higher Mg concentrations in HD trees, but the effect on LD trees is not significant. Keeping all other variables constant, with a 1 °C increase in mean temperature it is 6.6 times likely that HD trees will have higher Mg concentrations, while at the same time the likelihood of LD having higher concentrations is negligible (Table 5).

The results of the fitted MLR models indicate that beech on higher altitudes is less likely to have significant differences in N/P ratios and increasing maximum temperature has the same effect (Table 6). A similar effect of maximum temperatures can be seen for differences in N/Ca. Holding all other variables constant, with an increase in precipitation the odds are 0.6% higher that HD trees will have higher N/P ratios and 0.8% higher N/Mg ratios. When mean temperatures and soil pH increase, it is 2.5 and 3.7 times, respectively, more likely that LD trees will have higher N/Mg ratios. Keeping all other variables constant, with a unit increase in PDSI, i.e., less drought, the odds are 83.5% higher that LD trees will have higher N/Ca ratios. The fitted MLR models for N/K did not indicate a significant influence of any of the tested environmental factors on the occurrence probability of differences in foliar nutrient ratios.

## 3. Discussion

Assessing the nutritional status of trees using the values of element concentrations in needles or leaves is a common diagnostic practice in forestry [40,41,42]. In Croatia there have been only a few studies into the nutritional status of beech [41,43,44], however these studies were performed on a limited research area. The results obtained in this study indicate a generally satisfactory nutrition of common beech in Croatia. Mean concentrations on the study and plot levels were within the normal range for beech [40]. Similar results were reported by Ognjenović et al. [39] from a long-term monitoring based case study of beech foliar composition on one ICP Forests intensive monitoring plot.

Large deviations from the mean concentration values of longer periods in individual years have been reported for several mineral nutrients [45,46,47]. In contrast, foliar K concentrations were reported to be relatively stable [39], and this finding is repeated in our study, although concentrations of all investigated elements stayed mostly within limit values [40]. Very similar relationships of N and K to limit values has also been observed in other studies [48,49]. On the other hand, P foliar levels are a reason for concern at the European level [12,15]. While beech P nutrition on a study area level seemed to be sufficient [40], on average 33% of the plots had insufficient P nutrition during the study period (Appendix A). Additionally, the ratio of nitrogen to phosphorus shows that beech nutrition in 2019 and 2020 was not balanced in this respect, which is partly caused by diminishing P, and partly by increasing N concentrations, respectively.

The fact that we found surplus N/Mg ratios in 2019 and 2020, both due to increasing N and decreasing Mg values, echoes results of other studies that report an unbalanced Mg nutrition on national [48,50] and European level [12].

Excessive Ca concentrations on an extremely high proportion of plots (90%) were determined in a study of beech nutrition in northern Spain, but since no symptoms of unbalanced nutrition have been identified in the field, the authors believe that the deviations of the obtained values may be based on ecological and geological differences between the studied forests and central European forests on the basis of which reference values were obtained [51]. This consideration is consistent with the hypothesis that the optimal leaf composition of each species depends on the specific ecological biogeochemical niche that the species occupies [52]. Taking into account the wide ecological amplitude of common beech, there is a possibility that the optimal range of concentrations of nutrients is adapted to the specific habitat in which a particular stand develops. Thus Sardans and Peñuelas [53] assume that plant species have a certain degree of flexibility in changing stoichiometry in response to changes in environmental conditions such as climate gradients.

Nutritional status and tree crown defoliation, as indicators of vitality, can provide a basis for monitoring the effects of long-term environmental changes on forest ecosystems [22,48]. According to Simon and Wild [54], if the concentration of a certain element remains in the normal range, a decrease in mineral nutrition should be regarded more as a consequence than the cause of damage. If, on the other hand, the concentrations are inadequate, we can suspect nutrition to be the cause of damage. Although we did not find a universal relationship between the defoliation categories and nutritional status of beech trees, subsequent comparisons revealed differences in the element concentrations and ratios for certain years. For instance, we found significantly higher P concentrations in LD trees in year 2019. On the contrary, a study in northern Spain recorded higher P concentrations in more damaged trees [51]. The most pronounced differences in concentrations between defoliation categories were determined for Ca, with HD trees having significantly higher concentrations in 2018 and 2019. Higher concentrations in HD trees were also recorded for Mg in 2018. Conversely, Jonard et al. [35] report that higher defoliation values are associated with lower Ca and Mg concentrations in a liming and P/K fertilization experiment of beech stands in Belgium.

A case study investigating interrelations of various common beech vitality indicators did not find significant links between foliar nutrient concentrations and defoliation [39]. It should be taken into account that (i) the study was performed on a limited area, (ii) element concentrations were mostly within the normal range, and (iii) the range of defoliation values was narrow and seldom exceeded 25%. The lack of association between N concentrations and defoliation values, also recorded in this study, Thimonier et al. [36] attribute to the same kind of limitations. Unlike Ouimet and Moore [55], who observed that an increase in K concentration in needles resulted in a decrease in defoliation in balsam fir stands, we found no significant differences in K concentrations between defoliation categories. Compared to fertilization experiments, the analysis of the nutritional status in natural stands is more demanding due to the high variability of data and the inability to control many factors that may affect the nutritional status. Ewald [56] states heterogeneous environmental conditions as the reason for the lack of significant relationship between nutrient concentrations and spruce defoliation status in a study conducted in Bavaria. On the other hand, with this study we do have a full picture of beech nutritional status in Croatia, including the insight into the significant variations between years and defoliation categories. Significant differences in concentrations only during certain years are not a phenomenon specific to this research alone. High year-to-year variability in nutrient concentrations is characteristic of nutritional studies [15,35,48,57].

The quantification of forest ecosystem response in a climate driven changing environment is fundamental for maintenance, enhancement and restoration of future forest ecosystem goods and services. The results presented in this study are a novel approach in investigating the effects of climate and environmental properties on the defoliation/nutrition relationship. Both tree nutrition and defoliation have been reported to depend on climate properties in various studies. A study by Braun [58] showed that air temperature and precipitation have a considerable impact on the foliar nutrient concentration. Jonard et al. [49] observed a positive relationship between precipitation and foliar Ca concentrations, while insufficient water supply during dry spells have a negative impact on the uptake of calcium, according to Bergmann [59]. Our results indicate that the increase in precipitation is better utilized in low defoliation trees since the odds are 90% higher for LD trees to have higher Ca concentrations with an increase in precipitation. Lukac et al. [60] state that elements with primarily biologically controlled cycles such as nitrogen can show a different reaction to temperature changes than elements whose cycles are controlled by both biological and geological processes (P and K) or predominantly by geological processes (Ca and Mg). Therefore, it is not surprising that we did not determine a significant effect of environmental factors on the occurrence of differences in N concentrations between defoliation categories. A study of beech nutrient dynamics along a precipitation gradient found that N concentrations remain constant with decreasing precipitation while P concentrations increase, resulting in a decrease in N/P ratio [61]. However, in this study increasing precipitation increases the odds of higher N/P ratios in high defoliation trees suggesting a disturbed uptake and utilization of P in beech with higher defoliation.

From our results it becomes quite clear that an unequivocal association of foliar nutrition and crown status should not be expected for beech. It seems plausible that various pressures influence this relationship, affecting the physiological status of beech trees in a multitude of ways. For instance, Gottardini et al. [22] found that various morphological and physiological characteristics of beech leaves and canopy such as leaf volume and photosynthetic activity, had a significant negative association with the extent of damage. Additionally, Ferretti et al. [62] showed that the 25%-defoliation threshold can be a reasonable approximation for tree classification indicated by the effect of slight and moderate variations in defoliation on tree growth, which is in contrast to the results obtained by Tallieu [63]. Highly differing research results on this topic are likely depending on the research area, management practices, climate influences and other unknown factors. Nevertheless, we established that environmental factors, especially climate, influence the occurrence of differences in foliar nutrition between defoliation categories.

## 4. Materials and Methods

### 4.1. Study Area and Plot Design

The ICP Forests Level I monitoring plots in Croatia are established on intersections of a 16 × 16 km grid that contain forest cover. These plots do not have a fixed area; rather, 24 trees are chosen for defoliation assessments using a cross-cluster system with six trees in each cluster [64]. Only plots with a minimum of five beech trees in the sample in 2018 were selected to ensure that beech was significantly represented in the mixture of tree species, resulting in total of 28 plots (Figure 3).

### 4.2. Climate Data

Climate monitoring stations are generally situated at considerable distances from the forest research plots. Therefore, the data they provide are not always representative of the research locations. To overcome this, we used gridded data produced by regression kriging (RK), which is a hybrid method of interpolation carried out in four steps [68,69]. The method was validated with leave-one-out cross-validation, while the root mean square error (RMSE) was calculated between observed and interpolated values. Mean RMSEs are for mean monthly temperature from 0.5 °C to 0.9 °C, for minimum temperature from 1.1 °C to 1.5 °C, for maximum temperature from 0.7 °C to 1.1 °C, and for precipitation from 18 to 30 mm, averaged by months.

Monthly temperature and precipitation values from the gridded dataset on 1 km spatial resolution for Croatia [30] were used to calculate Mean Annual Temperature (MAT), mean annual minimum (Tmin) and mean annual maximum (Tmax) temperature, Mean Annual Precipitation (MAP) as well as the Palmer Drought Severity Index (PDSI) [70] and Standardized Precipitation Evapotranspiration Index (SPEI) [71]. Lower values of scPDSI and SPEI indicate a stronger drought intensity while higher values indicate a higher degree of humidity. SPEI was calculated on a time scale of 3 months.

### 4.3. Tree Selection Procedures, Foliar Sampling and Analysis

Sampling was conducted annually from 2018 to 2020 during late July and August. At a distance of less than 50 m from the centre of the plot, a stratified sampling of trees based on the percentage of defoliation was carried out, forming two groups of five trees: (i) “LOW DEFOLIATION” (LD) trees with defoliation lower than 25% and (ii) ‘HIGH DEFOLIATION’ (HD) trees with defoliation higher than 25%. Defoliation assessments were performed during July and August according to the ICP Forests methodology [24] in steps of 5%, ranging from 0 to 100%. An absolute reference tree, defined as the best/healthiest possible beech tree, was used in the assessments, regardless of local factors that may affect the crown condition. Only dominant or codominant trees were selected, without any wounds or fruiting bodies of fungi on the trunk. In cases when trees changed their defoliation percentage and thus changed their group status (LD/HD), new, replacement trees were selected instead, and the discarded trees were not used in further sampling. Stratified sampling allowed us to circumvent the irregular distribution of tree defoliation status on plots and focus on determining whether concentrations/contents of nutrients in leaves differ in trees of different defoliation categories on each plot. Leaves were sampled from the upper third of the crown using a shotgun rifle or a rope climbing technique [72,73]. Samples from each tree were combined into a composite sample according to fresh weight and analyzed in the Laboratory for Physical and Chemical Testing (LFKI) of the Croatian Forest Research Institute. Upon arrival to the laboratory, samples of plant material were dried at 105 ° C to constant mass, ground in a Fritsch Pulverisette 14 mill and prepared for analysis in a Milestone Ethos One microwave oven [74]. Concentrations of total nitrogen were determined on a Leco CNS 2000 elemental analyzer (LECO, 2000). Phosphorus (P) concentration was determined colorimetrically on a LaboMed UV/VIS spectrophotometer [74], and potassium, calcium and magnesium by atomic absorption on the Perkin-Elmer Aanalyst 700 absorption spectrophotometer [74].

### 4.4. Soil Sampling and Analysis

Soil sampling was performed in 2019 on five points located within each of the research plots. One point was located within each of the four clusters of trees that are assessed for defoliation during regular monitoring activities, and an additional fifth point was located in the centre of each plot. Soil samples were taken with a pedological drill from a depth of 0–10 cm, 10–20 cm, 20–40 cm, and 40–80 cm. Collected samples were pooled by sampling depth. Soil mechanical properties were determined by [75] and chemical parameters were analysed according to the following protocols and methods:soil pH (CaCl_2_) [76]exchangeable K, Ca and Mg according to [77]exchangeable acidity (free H^+^) according to [78]total N with an elemental analyzer Leco CNS 2000 [79]available P and K with the AL method [80]; P by spectrophotometry using molybdate blue method, on UV/VIS spectrophotometer LaboMed and K directly from filtrate on flame photometer Buck scientific PFP−7 [81].

### 4.5. Data Analysis

Descriptive statistics of foliar concentration and ratios were performed for each element. Foliar data distribution was inspected visually and by Shapiro-Wilk test. Slight deviations from normality were determined, especially for potassium, calcium and magnesium. However, studies have shown that slight deviations from normality do not affect the rate of false positive results [82].

Therefore, a two-factor repeated measures analysis of variance (ANOVA) was used to determine the difference in foliar concentrations and ratios between defoliation categories (LD/HD) of common beech on a study area level. The data of all elements met the assumption of homoscedasticity, which was confirmed by Levene’s test. The Holm–Bonferroni method [83] was used to perform pairwise comparisons to determine differences in foliar concentrations (i) between LD and HD categories in single years and (iii) across defoliation categories between sampling years.

To determine differences in foliar concentrations and ratios between defoliation categories on a plot level, we used the independent samples t-test [84]. The results were then categorized for each plot and sampling year to reflect three cases: (i) *HD_higher* if HD trees had significantly higher foliar concentrations (*p* < 0.05), (ii) *LD_higher* if LD trees had significantly higher foliar concentrations (*p* < 0.05) and (iii) *NO_diff* if there were no significant differences in foliar concentrations between defoliation categories, thus creating a variable which represents Differences in Foliar Concentrations or Ratios for each element (DFC/DFR).

Multinomial logistic regression (MLR) was used to determine which environmental factors influence the occurrence probability of a certain DFC/DFR category, except no_dif which was set as the reference category. Separate odds ratios, which are the exponentiation of model coefficients, were determined for all independent variables for each DFC/DFR category. The odds ratio of a coefficient indicates the occurrence probability of either HD or LD trees having significantly higher foliar concentrations over the probability of no difference between defoliation categories (*NO_diff*). An odds ratio > 1 indicates that the occurrence probability of a certain DFC/DFR category relative to the occurrence of *NO_diff* increases as the independent variable increases, whereas an odds ratio < 1 indicates that the occurrence of a certain DFC/DFR category decreases. Of the potential independent variables, those with the highest variable importance were selected with the random forest algorithm [85]. The final model selection process was based on diagnostic diagrams and a procedure defined by Johnson et al. [86]. All analyses were conducted in an R programming environment [87].

## 5. Conclusions

Our final view is that links between defoliation and nutrition of beech in Croatia are not very direct or very prominent; this can be seen from (I) the fact that differences in nutrition of LD and HD trees were found for only a few nutrients (II) these effects are mostly not universal, but present only in some years and on a limited number of plots and (III) nutrient concentrations mostly stay within normal values regardless of tree defoliation status. However, it is clear that environmental factors, especially climate properties, do affect the relationship between defoliation and nutrition. Discovering mechanisms by which environmental factors affect foliar nutrition should complement current research on the defoliation–nutrition relationships.

## Figures and Tables

**Figure 1 plants-12-00168-f001:**
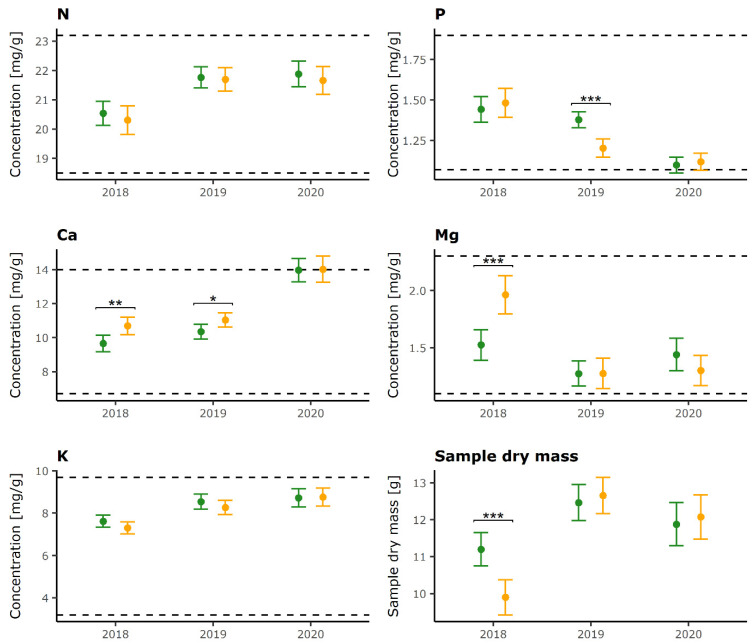
Mean foliar concentrations and sample dry mass (dots) with confidence intervals (vertical bars) for LD (green color) and HD trees (orange color) on all plots within sampling years. Dotted lines represent lower and upper critical value for normal range of foliar N concentrations for common beech according to Mellert and Goettlein [40]. Only significant differences in foliar concentrations between LD and HD categories determined by the Holm–Bonferroni method are indicated (* *p* < 0.05, ** *p* < 0.01, *** *p* < 0.001).

**Figure 2 plants-12-00168-f002:**
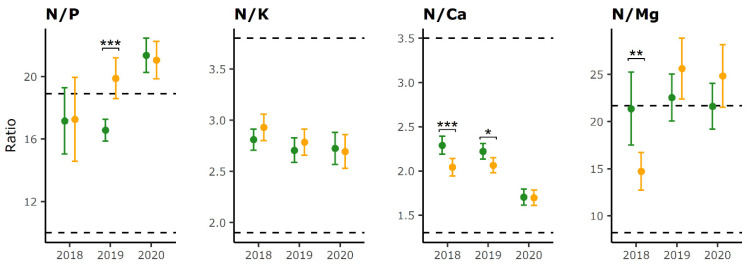
Mean foliar ratios (dots) and confidence intervals (vertical bars) for LD (green color) and HD trees (orange color) on all plots within sampling years. Dotted lines represent lower and upper critical value for normal range of foliar ratios for common beech according to Mellert and Goettlein [40]. Only significant differences in foliar ratios between LD and HD categories determined by the Holm–Bonferroni method are indicated (* *p* < 0.05, ** *p* < 0.01, *** *p* < 0.001).

**Figure 3 plants-12-00168-f003:**
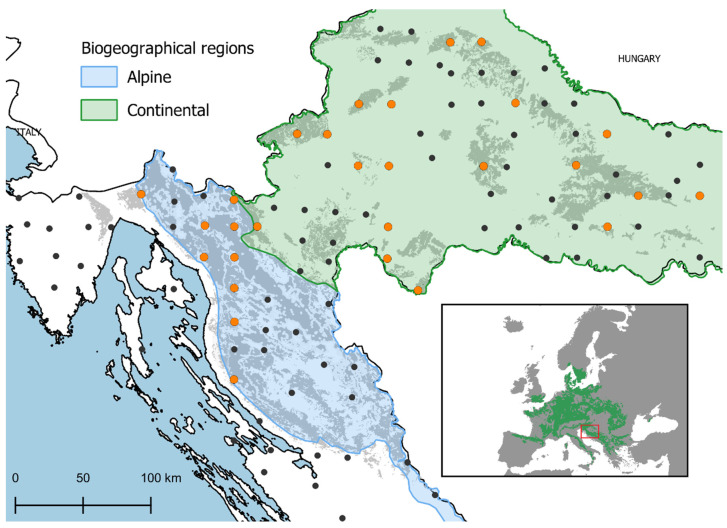
Location of research plots and distribution of European beech in Europe [65]. Selected Level I ICP Forests monitoring plots (orange dots); remaining Level I plots in Croatia (black dots); distribution of European beech forests in Croatia [66] (gray polygon); biogeographic region in the study area (shaded polygon) [67].

**Table 1 plants-12-00168-t001:** Results of Holm pairwise comparisons of foliar nutrient concentrations and sample dry mass within a defoliation category and between sampling years (NS–not significant, * *p* < 0.05, ** *p* < 0.01, *** *p* < 0.001).

Defoliation Category	Year 1	Year 2	N	P	K	Ca	Mg	Sample Dry Mass
LD	2018	2019	***	ns	***	**	***	***
LD	2018	2020	***	***	***	***	ns	*
LD	2019	2020	ns	***	ns	***	*	*
HD	2018	2019	***	***	***	ns	***	***
HD	2018	2020	***	***	***	***	***	***
HD	2019	2020	ns	*	**	***	ns	*

**Table 2 plants-12-00168-t002:** Number of plots with significantly higher foliar concentrations in each defoliation category within sampling years determined by the independent samples *t*-test (total number of plots: 28).

		N	P	K	Ca	Mg
2018	HD higher	1	-	-	6	4
	LD higher	1	-	2	-	1
2019	HD higher	-	-	1	-	2
	LD higher	1	4	1	1	1
2020	HD higher	-	2	1	1	1
	LD higher	2	-	-	-	1

**Table 3 plants-12-00168-t003:** Results of Holm pairwise comparisons of foliar nutrient ratios within a defoliation category and between sampling years (ns–not significant, * *p* < 0.05, *** *p* < 0.001).

Defoliation Category	Year 1	Year 2	N/P	N/K	N/Ca	N/Mg
**LD**	2018	2019	ns	ns	ns	ns
**LD**	2018	2020	***	ns	***	ns
**LD**	2019	2020	***	ns	***	ns
**HD**	2018	2019	ns	ns	ns	***
**HD**	2018	2020	*	*	***	***
**HD**	2019	2020	ns	ns	***	ns

**Table 4 plants-12-00168-t004:** Number of plots with significantly higher foliar ratios in each defoliation category within sampling years determined by the independent samples *t*-test (total number of plots: 28).

		N/P	N/Ca	N/Mg	N/K
**2018**	HD higher	-	-	1	2
	LD higher	-	5	2	-
**2019**	HD higher	5	1	1	2
	LD higher	-	2	1	1
**2020**	HD higher	1	-	-	-
	LD higher	1	2	1	-

**Table 5 plants-12-00168-t005:** Results of multinomial logistic regression for Differences in Foliar Concentrations (DFC) of potassium, calcium and magnesium and corresponding model AIC. Odds ratios are reported for each environmental factor while the confidence intervals are given in the brackets (* *p* < 0.1, ** *p* < 0.05, *** *p* < 0.01).

		DFC	
Element	Environmental Factor ^1^	HD Higher	LD Higher
**K**	**Alt**	1.000	1.005 ***
		(0.996, 1.003)	(1.002, 1.008)
	**MAT**	0.644 ***	0.884
		(0.480, 0.809)	(0.659, 1.108)
	**AIC**	48.042	48.042
**Ca**	**Tmax**	1.298 ***	0.00002 ***
		(1.112, 1.485)	(0.00002, 0.00002)
	**sand**	0.926	0.048 ***
		(0.809, 1.043)	(0.048, 0.048)
	**MAP**	0.988 **	1.900 ***
		(0.977, 0.999)	(1.900, 1.900)
	**Alt**	1.005 **	0.372 ***
		(1.001, 1.009)	(0.372, 0.372)
	**AIC**	56.908	56.908
**Mg**	**MAT**	6.578 ***	1.671 *
		(6.330, 6.826)	(1.149, 2.194)
	**MAP**	1.002	1.006
		(0.996, 1.007)	(0.998, 1.015)
	**Alt**	1.009 ***	1.001
		(1.005, 1.013)	(0.992, 1.009)
	**sand**	0.991	1.121 **
		(0.885, 1.097)	(1.020, 1.222)
	**pH**	2.600 **	1.015
		(1.708, 3.492)	(−0.815, 2.846)
	**AIC**	80.311	80.311

Environmental factors ^1^ Alt—altitude, MAT—Mean Annual Temperature, MAP—Mean Annual Precipitation, sand—soil sand fraction, pH—pH value, AIC—Akaike information criterion.

**Table 6 plants-12-00168-t006:** Results of multinomial logistic regression for Differences in Foliar Ratios (DFR) of potassium, calcium and magnesium. Odds ratios are reported for each environmental factor while the confidence intervals are given in the brackets (* *p* < 0.1, ** *p* < 0.05, *** *p* < 0.01).

		DFC	
	Environmental Factor ^1^	HD Higher	LD Higher
**N/P**	**MAP**	1.006 *	0.989
		(1.000, 1.012)	(0.967, 1.011)
	**Tmax**	0.820 ***	0.006 ***
		(0.696, 0.943)	(−0.438, 0.449)
	**Alt**	0.992 **	0.960 ***
		(0.986, 0.999)	(0.937, 0.983)
	**AIC**	59.378	59.378
**N/Ca**	**Tmax**	0.070 ***	0.402 ***
		(−1.553, 1.693)	(0.029, 0.775)
	**PDSI**	1.401	1.835 **
		(−0.956, 3.758)	(1.251, 2.419)
	**Ca^2+^**	0.534	0.909 **
		(−0.314, 1.381)	(0.829, 0.988)
	**MAP**	0.996	0.993 ***
		(0.957, 1.036)	(0.989, 0.998)
	**silt**	0.741	1.158 *
		(−0.353, 1.835)	(0.996, 1.319)
	**AIC**	72.131	72.131
**N/Mg**	**MAP**	0.999	1.008 ***
		(0.989, 1.009)	(1.002, 1.014)
	**MAT**	1.070	2.502 ***
		(0.648, 1.493)	(2.198, 2.805)
	**pH**	0.631	3.708 ***
		(−1.419, 2.681)	(2.770, 4.646)
	**SPEI**	2.613	0.071 **
		(−1.249, 6.474)	(−2.192, 2.333)
	**AIC**	65.799	65.799

Environmental factors ^1^ Alt—altitude, MAT—Mean Annual Temperature, Tmax—Mean annual maximum temperature, MAP—Mean Annual Precipitation, silt—soil silt fraction, SPEI—Standardised Precipitation-Evapotranspiration Index, PDSI—Palmer Drought Severity Index, Ca^2+^—Soil exchangeable calcium, AIC—Akaike information criterion.

## Data Availability

The data presented in this study are available on request from the corresponding author. The data are not publicly available due to legal reasons.

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
