# Peer review of "The Effect of Environmental Factors on the Nutrition of European Beech (Fagus sylvatica L.) Varies with Defoliation"

_plants, 2022, doi:10.3390/plants12010168_

Round 1

Reviewer 1 Report

At first, the manuscript have an interesting topic. This subject is of high interest.

However, many parts of the manuscript are formatted and structured in a perfunctory manner. For instance, all tables do have the same title “Results of Holm pairwise comparisons of foliar nutrient concentrations and sample dry mass within a defoliation category and between sampling years (NS –not significant, *p < 0.05, **p < 0.01, ***p < 0.001)”. This is not too funny, I believe. Due to this fact, I don't know what data are presented in the most number of tables. Then, I'm confused by the small sample size, represented by only 28 sample sites.

In addition, data of Fig. 1 and Fig. 2 show that values of the studied indicators are not stable, and they fluctuate considerably year by year without a well understandable trend. For instance, we see that in Fig. 2, the values of N/P are the same in 2018 between LD and HD, but in 2019 the HD value is significantly higher, while LD was the same as it was in 2018. Finally, in 2020, N/P values for both LD and HD became higher than in both 2018 and 2019, and they are almost equal (non-significant). Such pronounced fluctuations are highly questionable. I believe that the larger period of time is needed to make some conclusions under such instable results.

In Discussion, the authors are being confirmed my judgments by the sentence of “Although our sampling period was too short to derive meaningful trends, decreasing mean P concentrations can be observed in each successive sampling year”. Yes, of course, the study period can be relatively short. But you, authors, should explain why do you obtain such results. Otherwise, this is not a research, this is not an obtaining of new knowledge; this is just a fact statement, nothing more.

Another sentence in the Discussion states that "The results obtained in this study indicate a generally satisfactory nutrition of common beech in Croatia". But why do you think that this nutrition is satisfactory for the common beech there? There are no references to confirm this. In the Discussion, the authors don't discuss the environmental and site factors (see the title for this) leading to the obtained effects (concentrations of certain elements in leaves).
In addition, sometimes, we see speculative statements like in lines 289-290. If you didn't consider the fructification as one of factors, and there are no any data in the section Results, you cannot say about this factor and that it influences any elements (Ca and P in this case). I recommend to double-check the whole Discussion section and delete all speculative statements. Otherwise, include these data to the section Results or provide appropriate references.

In conclusion, I should say that the manuscript is needed in considerable revision connected with enlarging the sampling size, improvements in the style of the text, deleting speculative statements. As the best way, I would recommend to withdraw this manuscript and re-submit it, once the authors will have sufficient amount of data for such strong conclusions. If we say about leaf traits, I recommend to see the following paper https://dx.doi.org/10.24189/ncr.2022.016, which is based on 25-30 years of the research. This period is relatively suitable to make certain conclusions. But I understand that this my recommendation is not suitable for the authors. Therefore, I recommend to improve the manuscript according to my suggestions.

Reviewer 2 Report

The paper is worth of consideration, however I suggest some corrections.

The authors subdivided the plots into two average defoliation classes: > and < 25%, but they found only little and not significant differences. I think the 25% threshold may be not fully meaningful since there is no evidence on a possible physiological significance. Moreover, the authors do not show the distribution and the distance of mean values of defoliation for singular plots. Differences may be relevant with higher defoliation values occurring in a little number of plots. I think it should be more appropriate to consider singular plots for correlative statistical analyses.

It would be nice if also the values of defoliation in the three years are presented. In particular, I wonder if the 2018 megadrought in central Europe affected also Croatian forests. In this case, the differences between the sample dry mass in this year can be explicated.

Discussion is vague and speculative in several parts. For ex., “Atmospheric deposition of N”: do you have any data? “Significant number of plots”: how many? Do you perform statistical tests? Speculations on trends are not acceptable since the short period of assessment.

“Normal range” for nutrients content are not specified. Data on the nutritional status of beech leaves are available in literature. Why were they not considered as reference values?

Round 2

Reviewer 1 Report

After the manuscript revision, it looks much better than it was at the original submission. I thank you for your complete responses to my comments and suggestions. I am fully satisfied with answers and corrections made in the text.